# Availability of Guanitoxin in Water Samples Containing *Sphaerospermopsis torques-reginae* Cells Submitted to Dissolution Tests

**DOI:** 10.3390/ph13110402

**Published:** 2020-11-19

**Authors:** Kelly Afonsina Fernandes, Humberto Gomes Ferraz, Fanny Vereau, Ernani Pinto

**Affiliations:** 1Department of Clinical and Toxicological Analyses, Faculty of Pharmaceutical Sciences, University of São Paulo, Av. Prof. Lineu Prestes, 580, Butantã CEP 05508-900, São Paulo, Brazil; kelly.af@usp.br; 2Department of Pharmacy, Faculty of Pharmaceutical Sciences, University of São Paulo, Av. Prof. Lineu Prestes, 580, Butantã CEP 05508-900, São Paulo, Brazil; fyvereau@usp.br; 3Centre of Nuclear Energy in Agriculture, University of São Paulo, Av. Centenário, 303, Piracicaba CEP 13416-000, Brazil

**Keywords:** anatoxin-a(s), neurotoxins, cyanobacteria poisoning, bio-accessibility

## Abstract

Guanitoxin (GNT) is a potent neurotoxin produced by freshwater cyanobacteria that can cause the deaths of wild and domestic animals. Through reports of animal intoxication by cyanobacteria cells that produce GNT, this study aimed to investigate the bio-accessibility of GNT in simulated solutions of the gastrointestinal content in order to understand the process of toxicosis promoted by GNT in vivo. Dissolution tests were conducted with a mixture of *Sphaerospermopsis torques-reginae* (Cyanobacteria; Nostocales) cultures (30%) and gastrointestinal solutions with and without proteolytic enzymes (70%) at a temperature of 37 °C and rotation at 100 rpm for 2 h. The identification of GNT was performed by LC-QqQ-MS/MS through the transitions [M + H]^+^
*m/z* 253 > 58 and [M + H]^+^
*m/z* 253 > 159, which showed high concentrations of GNT in simulated gastric fluid solutions (*p*-value < 0.001) in comparison to simulated solutions of intestinal content. The gastric solution with pepsin promoted the stability of GNT (*p*-value < 0.05) compared to the simulated solution of gastric fluid at the same pH without the enzyme. However, the results showed that GNT is also available in intestinal fluids for a period of 2 h, and solutions containing the pancreatin enzyme influenced the bio-accessibility of the toxin more compared to the intestinal medium without enzyme (*p*-value < 0.05). Therefore, the bio-accessibility of the toxin must be considered both in the stomach and in the intestine, and may help in the diagnosis and prediction of exposure and risk in vivo through the oral ingestion of GNT-producing cyanobacteria cells.

## 1. Introduction

Guanitoxin (GNT) [1] (formerly Anatoxin-a(s)) is a potent natural neurotoxin produced by freshwater cyanobacteria [2,3]. Its mode of action is the same as synthetic organophosphates, in which the phosphate ester functional group binds to the active serine site of acetylcholinesterase (AChE), ultimately causing AChE block [3,4,5]. The result of the irreversible inactivation of AChE is the accumulation of acetylcholine in the synaptic clefts, thus causing cholinergic hyperstimulation that, in most cases, is lethal for organisms [6,7,8].

In the past, GNT has been associated with the death of domestic and wild animals that accidentally consumed water containing cyanobacterial cells [5,6,8,9]. The clinical signs observed in these animals consisted mainly of excessive salivation, muscle tremors, convulsions, fasciculation convulsions, and respiratory failure. The lethal dose (LD_50_) of GNT was determined in mice to comprise a range from 20 µg/kg to 40 µg/kg, with a survival time of 10 to 30 min, and it is considered ten times more toxic than other cyanotoxins of the same class [3,5,10]. Other LD_50_ and inhibitory concentration (IC_50_) values were observed in fish, Cladocera, and insects, showing symptoms of intoxication common to those observed in mammals [11,12,13,14,15].

There are no known variants of GNT; it is known that species of the genus *Dolichospermum* and *Sphaerospermopsis* are the main producers of this cyanotoxin [1,16]. AChE has been used as a biomarker to assess the presence of the toxin in aqueous samples [17,18]. However, enzymatic methods can generate false-positive results and can be influenced by the presence of synthetic organophosphates that are available in the environment. For this reason, analytical methodologies by LC-MS are more indicated due to the high specificity and sensitivity that they provide for the correct identification of GNT [19].

Although the presence of this toxin is less common than other cyanotoxins such as microcystins, extremely high levels of GNT have already been detected in water samples [9,20]. Furthermore, there are recent reports of the occurrence of this toxin through cases of accidental poisoning in dogs, who after drinking water with cyanobacterial cells, showed clinical signs of acute intoxication characteristic of GNT [6,21,22]. However, the monitoring of GNT in bodies of water for human use is not yet mandatory, and there are no limits for the detection of GNT established by the World Health Organization (WHO). The lack of consistent toxicological data and an analytical standard for quantifying GNT are the main factors that limit the mandatory monitoring of GNT in water bodies [18,23,24].

Environmental factors also imply a lack of data on the occurrence of GNT in water bodies, such as the instability of the toxin at high temperatures and slightly alkaline pH [7,25]. However, there are contradictory results regarding the time of degradation of the molecule, and it is not known whether GNT has resistance to other chemical substances. On the other hand, the occurrence of GNT has been reported in eutrophic environments with a slightly alkaline pH [26,27], which is associated with the predominance of species producing GNT. There is no precise information on the toxin’s half-life in the environment. However, even if it prevails in the environment for a short time, it can be sufficiently lethal depending on the available concentrations; it can cause severe impacts on aquatic and terrestrial biota.

Therefore, our work aimed to investigate the availability of GNT in simulated solutions of gastric and intestinal contents with and without digestive enzymes through in vitro tests, following guidelines established by the United States Pharmacopeia [28,29]. From the dissolution tests, we expect to provide information on the bio-accessibility of GNT in the gastrointestinal system in vivo, especially for wild and domestic animals that are generally the most affected by the toxic cyanobacteria available in eutrophic environments.

## 2. Results

The results presented in this study came from cultures of *Sphaerospermopsis torques-reginae* (ITEP-24), with a cell concentration of 3.29 × 10^6^. From dissolution tests, we obtained the profile of the accessibility of GNT in artificial solutions gastrointestinal content. The bio-accessibility of the toxin was measured by high-performance liquid chromatography-tandem triple-quadrupole mass spectrometry (HPLC-QqQ-MS/MS) with multiple reaction monitoring (MRM) using the transitions [M + H] + *m/z* 253 > 58 and *m/z* 253 > 159 (Scheme 1). Figure 1 refers to the total ion chromatogram and MRM for the GNT molecule, showing that the relative area of the GNT peak is greater in samples with acidic pH.

The results also showed that the GNT contents were statistically higher (*p*-value < 0.001) in treatments with simulated stomach contents (with and without enzyme) compared to treatments with solutions of simulated intestinal content with enzyme and without pancreatin. Then, we compared the contents of GNT between solutions with the same pH. The graphs shown in Figure 2 show the contents (%) of GNT while the data presented in Appendix A represent the raw data obtained by LC-MS.

Treatment with the enzyme pepsin (pH 1.2) showed that the concentration of GNT increased significantly from zero to 10 min (*p*-value < 0.0001) and then continued to increase slowly, with a tendency to stabilize the concentration of the toxin until the end of the experiment. In addition, the Tukey test showed that the concentration of GNT at 10 min was statistically different from all times, except 5 and 15 min. From 0 to 15 min was the period of greatest activity of the pepsin enzyme acting on cell breakdown of the ITEP-24 strain, and consequently, availability of GNT in the extracellular medium (Figure 2A).

We compared the results obtained from the simulated fluid dissolution test of the gastric contents with the pepsin enzyme (pH 1.2) and without (pH 1.2); the statistical analysis shows that the GNT concentration at 20 min were statistically equal in the two treatments (Figure 2A). However, while the concentration of GNT begins to increase initially (0–15 min) (*p*-value < 0.0001) in the treatment with the enzyme, the opposite occured in treatments with gastric solution without the enzyme (*p*-value < 0.0001) (Figure 2A).

Figure 2B shows the results obtained in the treatments with the simulated solution of the intestinal content with and without the pancreatin enzyme. In the dissolution test performed with the simulated solution of intestinal fluid with the pancreatin enzyme, the concentration of GNT initially increased significantly between 10 to 15 min, and was statistically higher compared to all other periods (*p*-value < 0.0001). After 15 min, the concentration drastically decreased to 20 min, maintaining stability for up to 30 min, and from 30 min to 60 min, there was a slight increase in the concentration of the toxin, which then tended to stabilize until the end of the experiment. In treatments without the pancreatin enzyme, the toxin content released was greater 0–10 min (*p*-value < 0.05) and from 15 min to 120 min there was no statistical difference.

The intestinal solution treatments with the pancreatin enzyme (pH 7.5) were compared with the results obtained in the test with the simulated solution of the intestinal content without the enzyme. The statistical tests showed significant differences in the times of 10–15 min (*p*-value < 0.0001) and 90–120 min (*p*-value < 0.05). For all other times, there was no statistical difference (Figure 2B).

At the end of the experiments, measurements of the pH value were performed in all dissolution test solutions with no significant changes in the acidic (1.20 to 1.33) and alkaline (7.50 to 7.56) solutions. Microscopic analyses of the samples resulting from the dissolution tests were also performed to assess the rupture of the cell wall of the ITEP-24 strain. Microscopic analysis revealed a small number of isolated cells that appeared intact, and quite a few fragmented cells.

## 3. Discussion

The dissolution test was developed to study the time required for the drug to solubilize in humans and animals [29]. The dissolution method is also used to evaluate and guarantee the quality of the drugs made available for use, in which it is established that at least 80% of solid formulations are dissolved in a short period [30,31]. In general, this method aims to predict whether solid formulations will be dissolved in a simulated aqueous medium of the stomach and intestine, leaving the active ingredient available to be absorbed into the bloodstream to exert its effect on the individual [32]. In vitro dissolution tests are usually used before in vivo tests [33]; therefore, this method can be explored to assess the bio-accessibility of several active compounds of cyanobacteria of pharmaceutical and food interest [34,35,36].

In this sense, this study aimed to investigate the amount of intracellular GNT released in simulated media from the gastrointestinal tract when subjected to controlled experimental conditions. Based on the principles of the dissolution method, the solubility of GNT in acidic (pH 1.2) and basic (pH 7.5) media was evaluated in order to understand the bio-accessibility of GNT in the gastrointestinal tract in vivo due to cases of intoxication of animals by this cyanotoxin, as reported in the literature. The results presented here showed the relevant content and greater stability of GNT in simulated solutions of gastrointestinal content with and without enzyme action. On the other hand, the release of the toxin was more significant in solutions of stomach contents, and even greater in solutions with the presence of the enzyme pepsin.

In the simulated gastric fluid dissolution test without enzymes, although the concentration of GNT was also relevant, there was a drop in concentration after 10 min; this can be justified by the instability of the toxin in acidic solutions with pH values below 3.0. Studies carried out to evaluate the GNT molecule’s stability at different pH and temperatures showed that the toxin is stable at pH 3.0 and can be slightly degraded at acidic pH (pH 1.5 and 5.0) (unpublished results).

Dissolution tests with proteolytic enzymes are normally used for formulations coated with some gelatinous membrane that does not dissolve in an aqueous medium due to crosslinking which can interfere with the dissolution of the drug [37]. In the case of our study, GNT was protected by cyanobacterial cells. Cyanobacteria have a cell wall made up of peptidoglycan, proteins, and lipopolysaccharides, which have protective and compliance functions [38,39]. In addition, the cell wall forms a boundary between cell constituents; therefore, some stimulation is needed to break the cell wall to release intracellular components of interest [40].

According to the dissolution method’s conditions, the use of the enzymes pepsin and pancreatin, together with the constant rotation promoted by the rotating blades at 100 rpm, could act in the cell lysis of cyanobacteria. In studies of extraction of secondary cyanobacteria metabolites, organic solvents and suitable equipment are usually used, such as ultrasound probes or thermal shock of cells by freezing and thawing; these methods assist in cell breakdown, promoting the release of intracellular metabolites [40].

The results of this study showed that the concentration of GNT in the dissolution test with the pepsin enzyme had an exponential growth character from 5 to 20 min; the opposite occurred in the dissolution test with simulated gastric fluid (Figure 2A). Pepsin probably acted in the breakdown of proteins present in the cyanobacterium cell wall, thus enabling the continuous release of GNT in the extracellular medium. Pepsin acts on protein metabolism by transforming them into simpler peptides; it catalyzes the hydrolysis of peptide bonds adjacent to amino acids with side chains, aromatic amino acids (phenylalanine, tryptophan and tyrosine), branched chain amino acids, and methionine [41]. Pepsin may also have had a greater role in degrading other metabolites present in the ITEP-24 strain, thus making GNT more available. However, these hypotheses require more specific studies to answer questions regarding the interaction of GNT with other molecules, especially digestive molecules.

In dissolution tests with simulated solutions of stomach fluid, usually lasting 1 h, the time was extended to 2 h to compare the tests performed with simulated solutions of intestinal content. The data show that higher GNT concentrations were more available in simulated stomach fluid solutions than simulated intestinal fluid solutions. However, although the concentration of GNT was lower in dissolution tests with simulated fluids from the intestine, the results showed that GNT was not fully degraded in the 2 h period. The fact is that some studies talk about the degradation of the toxin in alkaline solutions, but this is not entirely correct. Therefore, based on this study, we can understand that most of the toxin is available in the stomach, and can also reach the intestinal tract.

In the dissolution test performed with the pancreatin enzyme, there was an increase in the concentration of GNT from 5 to 15 min, which was statistically significant (*p*-value < 0.0001) compared to the results of the intestinal solution (Figure 2B). The pancreatin used in the dissolution assay comprises of a mixture of several enzymes including trypsin, amylase, lipase, ribonuclease, and protease. Pancreatic enzymes such as trypsin act mainly in the hydrolysis of lysine and arginine esters [42]. Thus, this set of pancreatic enzymes may have acted in the breakdown in the cell wall of the ITEP-24 strain, allowing the release of GNT into the extracellular environment; however, GNT is not very stable in a basic environment, so high concentrations of GNT may have been released and then hydrolyzed.

Therefore, as demonstrated in this study, the low concentration of the toxin at pH 7.5 may be associated with the low instability of the GNT at pH > 7.0. However, although its concentration was lower in tests with artificial solutions of the intestinal system, concentrations of the toxin remained in the solutions for a period of up to 2 h. The remains of cells that were still intact after 2 h indicated that gastrointestinal content solutions with and without enzymes managed to release only a part of the intracellular toxin. Depending on the morphology and metabolism of the digestive system in vivo, the presence of intact cells after 2 h can result in the continuous release of GNT in the body and cause prolonged toxic effects, even at a low concentration.

Cyanobacteria produce a variety of secondary metabolites that are toxic to many organisms [43,44,45,46], including man [47]. Most GNT-producing cyanobacteria have been identified in freshwater environments accompanied by animal poisoning. Birds and domestic mammals were the groups most affected after water consumption containing cyanobacterial cells that produce GNT [6,9,22]. The data available in the literature also show that the effect of this cyanotoxin toxicity is very rapid, affecting the nervous system to cause seizures and muscle paralysis, followed by sudden death [6,8,9,22].

In addition, laboratory tests carried out on mice and rats using intraperitoneal injections containing GNT showed that the toxic effects are more pronounced between 7 to 30 min and can cause rapid death in up to 60 min [5,6,7,9]. The results presented here showed higher concentrations of GNT in the dissolution tests carried out with the enzymes pepsin and pancreatin in the initial phase of the experiments (10 and 20 min), corroborating tests in vivo where the toxic signs were more severe in the same period.

Tests carried out on other animals treated with crude extracts of cyanobacteria producing GNT showed that the toxic effect of GNT can vary according to the method of exposure and the dose administered [8,11,48,49]. However, both tests with the pure toxin and tests that used cells containing the toxin showed similar clinical effects, and in many cases was followed by death of the organisms.

Despite several studies showing the negative effect of GNT on aquatic and terrestrial biota [6,8,50], the monitoring of this potent neurotoxin is not yet routinely performed in bodies of water by regulatory agencies and there is no standardization of limits for the concentration of GNT. However, there is a particular concern with aquatic organisms, which may share the same habitat with toxin-producing cyanobacterial species. In addition, cattle, domestic animals such as dogs as well as water birds can be exposure to contaminated water during bloom events. We believe that our findings can contribute substantially with information about GNT released from cyanobacterial cells after ingestion of contaminated water mainly by mammals. This in vitro released pattern can be useful for comparison in real cases of GNT poisoning and treatment of animals.

## 4. Materials and Methods

### 4.1. Reagents

The reagents employed to prepare the Artificial Seawater Medium (ASM-1) used to grow the cultures of the ITEP-24 strain were of analytical grade (Vetec, Rio de Janeiro, Brazil); the Lugol used to fix the culture of the ITEP-24 strain was purchased from Sigma-Aldrich (Sigma-Aldrich, Darmstadt, Germany). Reagents employed to prepare mobile phases used in LC-MS analyses were MS grade Acetonitrile, formic acid, and ammonium formate were all purchased from Sigma-Aldrich (Sigma-Aldrich, Darmstadt, Germany). The reagents used to prepare simulated solutions of gastric and intestinal fluids were sodium hydroxide, sodium chloride, hydrochloric acid (purity over 99%) (Sigma-Aldrich, Darmstadt, Germany), monobasic potassium phosphate, monobasic sodium phosphate, and phosphoric acid (Vetec, Rio de Janeiro, Brazil).

The enzymes used in the dissolution tests were pepsin from lyophilized powder from porcine gastric mucosa (Sigma P7000, ≥250 units/mg) and pancreatin from pig stomach and pancreatin from porcine pancreas (Sigma P-1500 4X United States Pharmacopoeia (USP) specifications) containing enzymatic components including trypsin, amylase, lipase, ribonuclease and protease, produced by the exocrine cells of the swine pancreas (Sigma-Aldrich, Darmstadt, Germany). Buffer solutions had pH values of 7.00, 4.00 and 9.00 (Millipore, Milford, MA, USA). Ultrapure water used to prepare all solutions was obtained from a Direct-Q8 water purification system (Millipore, Milford, MA, USA).

### 4.2. Cultivation of the Sphaerospermopsis torques-reginae (ITEP-24) Strain Producing Guanitoxin

The dissolution tests were performed with cells of the species *S. torques-reginae* (Komárek) [16], obtained from the Technological Institute of Pernambuco (ITEP). The strain ITEP-24, classified as a GNT producer [19], was isolated from water samples collected in the Tapacurá/PE/Brazil reservoir [26]. Currently, this strain is also part of the collection of cyanobacteria of the Laboratory of Toxins and Natural Products of Algae and Cyanobacteria of the Faculty of Pharmaceutical Sciences of the University of São Paulo/Brazil (FCF/USP).

The ITEP-24 strain was grown in ASM-1 medium [51] at pH 7.5–8.0 with continuous aeration for a period of approximately 20 days. During this period, the cultures were maintained at a temperature of 22.0 ± 1.0 °C and a 12:12 photoperiod (Nova Técnica, São Paulo, Brazil) under a light intensity of 40 μmol of photons m^−2^ s^−1^ measured by a quantum sensor (QSL-100, Biospherical Instruments Inc., San Diego, CA, USA).

After reaching exponential growth, the cultures were centrifuged at 15,000× *g* at 4 °C for 10 min, (Eppendorf 5804R centrifuge, Eppendorf AG, Hamburg, Germany). Before the dissolution experiments, 5 mL of the ITEP-24 strain culture was collected and fixed with Lugol. The samples fixed in Lugol were then inserted into a Neubauer Chamber (hemocytometer) for cell counting using a Zeiss Axiovert 135M optical microscope (Carl Zeiss, Göttingen, Germany). Cell counting was performed based on the number of cells per strand and cell density was performed according to Blakefield and Harris (1994) [52].

### 4.3. Preparation of Simulated Gastrointestinal Fluid Solutions with and without Digestive Enzymes

Dissolution tests were designed to understand how GNT would react after being released from *S. torques-reginae* cells (ITEP-24) in simulated solutions of gastric and enteric fluid with and without the presence of enzymes that help in the processes of metabolism of the stomach and intestine. The solutions used in the dissolution tests were prepared according to the United States Pharmacopoeia standards [28].

The simulated gastric fluid was prepared with 2.0 g of sodium chloride dissolved in ultrapure water, and 6.0 mL of hydrochloric acid was then added with ultrapure water to a volume of 1000 mL. The pH was adjusted with 1 M sodium hydroxide or 1 M hydrochloric acid until a pH of 1.2 was achieved. For the medium with the enzyme pepsin, the same solution was prepared (simulated gastric fluid), and 3.2 g of pepsin was added for a total volume of 1000 mL.

The intestinal solution was prepared with 6.81 g of monobasic potassium phosphate and 1.70 g of sodium hydroxide dissolved with ultrapure water to the volume of 1000 mL. The pH was adjusted with 1 M sodium hydroxide or 1 M phosphoric acid until a pH of 7.5 was achieved. The same medium (simulated intestinal fluid) was subsequently used to prepare the solution with the pancreatin enzyme, using 2.5 g of pancreatin for a total volume of 1000 mL of the simulated solution of the intestinal contents. The pH values were measured with a digital pH meter combined with a glass electrode (827 pH Lab/6.0224.100, Metrohm, Herisau, Switzerland). The pH meter was calibrated by standard buffer solutions with pH 7.00, 4.00 and 9.00.

### 4.4. Dissolution Test with the ITEP-24 Strain

The tests were carried out in a 708-DS dissolution apparatus (Agilent Technologies, Santa Clara, USA) equipped with 6 glass cylinders with a capacity of 1000 mL, using the rowing method at 100 rpm at a temperature of 37 ± 1 °C [53]. The experiments were carried out in two stages. First, experiments were performed with simulated stomach fluid solutions with and without the pepsin enzyme simultaneously, using three cylinders for each type of solution, i.e., each test was performed in triplicate. Each cylinder was composed of 700 mL of simulated gastric medium and 300 mL of culture containing cells of the ITEP-24 strain. For the simulated gastric solution with the enzyme pepsin, 2.24 g of the enzyme (560,000 U) was used; each mg of pepsin was equivalent to 250 U.

In the second stage, immediately after the tests with gastric solutions at pH 1.2, the experiments were carried out with simulated solutions of the enteric liquid with and without the pancreatic enzyme. The experimental conditions were the same mentioned in the previous experiment, i.e., the assays with simulated enteric solutions were performed together, representing three cylinders for each assay, each cylinder containing 700 mL of simulated enteric medium plus 300 mL of culture, representing a final volume of 1000 mL of medium. In the case of the pancreatin enzyme assay, a volume of 700 mL of intestinal solution with 1.75 g of the enzyme was used for each cylinder (>175,000 U of amylase activity, >14,000 U of lipase activity, and >175,000 U of protease activity). Each mg of pancreatin was equivalent to >100 U of amylase activity, >8.0 U of lipase activity, and >100 U of protease activity.

The dissolution tests lasted 2 h (120 min) and the samples were obtained through manual collections with 5 mL syringes connected to cannulas. The collections were performed after 5, 10, 15, 20, 30, 45, 60, 90, and 120 min, with total of 3 mL of solution for each time point (triplicate). After collection, the samples were filtered with a 0.45 µm Poly(vinylidene fluoride) (PVDF) membrane (Nova Analítica, São Paulo, Brazil) and stored on dry ice until the time of analysis in the LC-MS.

### 4.5. Liquid Chromatography-Tandem Mass Spectrometry

The samples referring to all extraction protocols were analyzed by high-performance liquid chromatographytandem coupled to a triple-quadrupole mass spectrometer (HPLC-QqQ/MS/MS) Agilent 6460 (Agilent Technologies, Santa Clara, CA, USA) with ionization by electrospray, in positive mode at 3500 V. Nitrogen was used as the gas nebulizer (45 psi) and drying gas (5 mL/min at 300 °C).

The separation of compounds was performed on a hydrophilic chromatographic column ZIC-HILIC, 150 × 2.0 mm, 5 µm (Merck, Darmstadt, Germany). The mobile phases consisted of A (water containing 10 mM ammonium formate and 0.04% formic acid) and B (acetonitrile/water (80:20 *v/v*), containing 5 mM ammonium formate and 0.01% formic acid) [19]. The injection volume was 5 µL, and the chromatographic separation was carried out in a linear gradient with a flow of 0.150 mL/min.

The toxin was identified using a linear gradient under the following conditions: from 0 to 10 min, the gradient was 90% (B). The mobile phase was then reduced to 40% (B) from 10 to 12 min, and from 12 to 12.5 min, the mobile phase was changed again to 90% (B). From 12.5 to 20 min, the mobile phase was maintained at 90% (B), finishing the analysis. The results were obtained by multiple reaction monitoring (MRM) and the identification of GNT was performed by the retention time, *m/z* 253 ([GNT + H]) and the quantifier and qualifier ions [M + H]^+^
*m/z* 253 > 58 and [M + H]^+^
*m/z* 253 > 159. Data analysis was performed using MassHunter Qualitative Analysis B06.00 software (Agilent Technologies, Santa Clara, CA, USA).

### 4.6. Statistical Analysis

The data obtained in this study were presented as mean ± standard deviation (SD). Significant differences were assessed by two-way ANOVA and Tukey’s test for multiple comparisons (*p*-value < 0.05 and *p*-value < 0.0001). The percentages expressed in the graphs was obtained after normalizing the data. All statistical tests were performed using R Statistical software version 3.1.2 [54] and graphs were made using Prism 7 (GraphPad Software, San Diego, CA, USA).

## 5. Conclusions

LC-MS analyses showed that GNT was statistically more stable in simulated solutions of the stomach component, and the presence of the enzyme pepsin resulted in greater stability of the toxin in acidic solutions. Although GNT is more stable in acidic solutions, such as stomach fluid, this study showed that GNT is also available in alkaline solutions (pH 7.5), mainly in the presence of pancreatic enzymes and, therefore, its availability must be considered both in the stomach and in the intestine. Therefore, the data presented here can be useful in the diagnosis and treatment of animals or humans affected by the accidental ingestion of cyanobacterial cells that produce GNT.

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
