# Peer review of "Availability of Guanitoxin in Water Samples Containing *Sphaerospermopsis torques-reginae* Cells Submitted to Dissolution Tests"

_pharmaceuticals, 2020, doi:10.3390/ph13110402_

Round 1

Reviewer 1 Report

This manuscript (MS) authored by Fernandes et al aimed at using dissolution assays to predict accidental ingestion of contaminated water with cyanobacterial bloom containing guanitoxin. The factors under scrutiny were the amounts and proportion of guanitoxin from intact cells of cyanobacteria Sphaerospermopsis torques-reginae determined by LC-MS QqQ and QTOF.

This research is interesting and worthy for publication. The manuscript is clear, well structured and easy to follow.

Minor Concerns:

Introduction. A paragraph should be added in this section dealing with the importance of cyanobacteria in different environmental, food and pharmaceutical applications. As for environmental applications, the authors may refer to the following articles: 10.1016/j.bej.2008.08.009, 10.1016/j.biortech.2010.01.054, 10.1016/j.apenergy.2011.11.019, 10.1016/j.jhazmat.2012.03.022

L338, the MS presents lots of information about guanitoxin toxicity, however, in the conclusion, authors did not mention the risk assessment of this cyanotoxin. I recommend authors include at least a sentence mentioning risk as they did in the abstract (from L31 to L33).

L255. Sphaerospermopsis torques-reginae was already abbreviated, changed to S. torques-reginae

L186, L191, L310 and L312: authors should standardize either 2 h or 120 min.

Figures and table are in good quality, but some of them need more resolution and should be changed (graphical abstract and fig 2 for instance). Some words are misspelled and grammar needs to be rechecked, therefore, I recommend careful revision by either a native English speaker or a professional language editing service to improve it.

To finish this revision, besides these above mentioned suggestions, the MS is good. This work is an important contribution to our knowledge of guanitoxin, its stability and behavior in dissolution studies as well as intercorrelation with cases of intoxication by contaminated water.

Author Response

São Paulo, October 30th, 2020

Authors’ responses - Referrers – Pharmaceuticals

Manuscript: “Availability of Guanitoxin in water samples containing Sphaerospermopsis torques-reginae cells submitted to dissolution tests”.

Authors’ answer: We would like to thank the Editor for your comments and detailed revision of our manuscript. All the suggestions pointed out have been accepted and were very helpful to improve the manuscript, making it clearer to the reader.

Open Review 1

Comments and Suggestions for Authors

This manuscript (MS) authored by Fernandes et al aimed at using dissolution assays to predict accidental ingestion of contaminated water with cyanobacterial bloom containing guanitoxin. The factors under scrutiny were the amounts and proportion of guanitoxin from intact cells of cyanobacteria Sphaerospermopsis torques-reginae determined by LC-MS QqQ and QTOF.

This research is interesting and worthy for publication. The manuscript is clear, well-structured and easy to follow.

Authors' response: We would like to thank you for your comments and suggestions for improving our work.

Minor Concerns:

#Introduction. A paragraph should be added in this section dealing with the importance of cyanobacteria in different environmental, food and pharmaceutical applications. As for environmental applications, the authors may refer to the following articles: 10.1016/j.bej.2008.08.009, 10.1016/j.biortech.2010.01.054, 10.1016/j.apenergy.2011.11.019, 10.1016/j.jhazmat.2012.03.022.

Authors' response: Thank you for the comment, we have inserted this information in the discussion, but our article covers more the pharmaceutical area, we quote the work proposed by Rodrigues et al. 2010 as suggested above.

Rodrigues, M. S.; Ferreira, L. S.; Converti, A.; Sato, S.; Carvalho, J. C. M. Fed-batch cultivation of Arthrospira (Spirulina) platensis: potassium nitrate and ammonium chloride as simultaneous nitrogen sources. Bioresour. Technol. 2010, 101, 4491—4498, doi:10.1016/j.biortech.2010.01.054.

#L338, the MS presents lots of information about guanitoxin toxicity, however, in the conclusion, authors did not mention the risk assessment of this cyanotoxin. I recommend authors include at least a sentence mentioning risk as they did in the abstract (from L31 to L33).

Dissolution tests showed cells of the intact ITEP-24 strain in contact with simulated gastrointestinal fluids could release significant concentrations of the GNT toxin.

Authors' response:  thank you for the observation; we have changed the text of the conclusion item according to your suggestion.

#L255. Sphaerospermopsis torques-reginae was already abbreviated, changed to S. torques-reginae

Authors' response:  Thank you for the observation, we have made the correction in the text.

#L186, L191, L310 and L312: authors should standardize either 2 h or 120 min.

Authors' response:  Thank you for the observation, we have standardized for 2 h.

Figures and table are in good quality, but some of them need more resolution and should be changed (graphical abstract and fig 2 for instance). Some words are misspelled and grammar needs to be rechecked, therefore, I recommend careful revision by either a native English speaker or a professional language editing service to improve it.

Authors' response:  We have improved the resolution of the graphical abstract figure and Figure 2. A native English speaker had already corrected the text, but we have made a new correction of the text, we hope that now it is in accordance with the grammar of the English language (please, see the newest certificate attached to the cover letter).

# To finish this revision, besides these above mentioned suggestions, the MS is good. This work is an important contribution to our knowledge of guanitoxin, its stability and behavior in dissolution studies as well as intercorrelation with cases of intoxication by contaminated water.

Authors' response:  We appreciate all comments and considerations, all of which were essential to improve our study.

Reviewer 2 Report

The manuscript by Fernandes et al. describes bio-accessibility of potent neurotoxin guanitoxin in simulated gastric and intestinal fluid. Using LC-MRM, authors quantitate guanitoxin levels in the two fluids over a course of 2 hours. They further investigate impact of adding digestive enzymes such as pepsin and pancreatin on availability of guanitoxin.

The study is well designed but leaves few important details that are needed for other researchers to successfully implement findings from this study.

  1. Authors have not described extraction procedure for the samples collected at different times of the dissolution test prior to LC-MRM analysis.
  2. Figure 2. It is not clear how authors calculated %GNT release. Please provide adequate details either in the main text or in the methods section.
  3. Figure 2B. For intestinal fluid the first data point coincides with 0 min. Please explain this discrepancy?
  4. Figure 2A. How do the authors explain mean guanitoxin release of 0% at time point 5 minutes in the gastric fluid?
  5. Why did the authors select pH of 7.5 and not pH 6.8 or try multiple pHs to simulate intestinal fluid from different segments of the intestine?
  6. While the authors mention they used equal volume of culture for the dissolution test, how did they ensure equal culture volume also equates to equal number of IETP cells per condition. Provide details in the method.

Author Response

São Paulo, October 30th, 2020

Authors’ responses - Referrers – Pharmaceuticals

Manuscript: “Availability of Guanitoxin in water samples containing Sphaerospermopsis torques-reginae cells submitted to dissolution tests”.

Open Review 2

Comments and Suggestions for Authors

The manuscript by Fernandes et al. describes bio-accessibility of potent neurotoxin guanitoxin in simulated gastric and intestinal fluid. Using LC-MRM, authors quantitate guanitoxin levels in the two fluids over a course of 2 hours. They further investigate impact of adding digestive enzymes such as pepsin and pancreatin on availability of guanitoxin.

The study is well designed but leaves few important details that are needed for other researchers to successfully implement findings from this study.

Authors' response: We would like to thank you for the comment and we will make all the adjustments in the text according to your suggestions.

#Authors have not described extraction procedure for the samples collected at different times of the dissolution test prior to LC-MRM analysis.

Authors' response:  thank you very much for the comment; we did not really add this data. However, guanitoxin is highly soluble in water with an octanol/water partition coefficient (CLogP) of -1.902. In addition, the objective of this study was precisely to evaluate the release of intracellular toxin in simulated media of gastrointestinal content and to evaluate how much toxin would be accessible in the body.

#Figure 2. It is not clear how authors calculated percentage GNT release. Please provide adequate details either in the main text or in the methods section.

Authors' response:  thank you for the comment. The data obtained by the LC-MS analyzes were then normalized. We will emphasize this in item 4.6., including statistical analyses. We believe that with this information in the text, it will be clearer for a better understanding by the reader.

#3Figure 2B. For intestinal fluid, the first data point coincides with 0 min. Please explain this discrepancy?

Authors' response:  thank you for the comment, we have changed the format of the figure, the scale of the X-axis has been modified to better understand the data presented.

#Figure 2A. How do the authors explain mean guanitoxin release of 0% at time point 5 minutes in the gastric fluid?

Authors' response:  The data were normalized using graphic prism software, where all curves start at 0 % and stabilize at 100 %, so the data was displayed in this way. However, after your observation, we agree that this is not the best way to show our results. For this reason, we have modified the graphics.

#Why did the authors select pH of 7.5 and not pH 6.8 or try multiple pHs to simulate intestinal fluid from different segments of the intestine?

Authors' response: thank you for the comment. We only tested pH 7.5, as recommended by the United States pharmacopoeia[1,2]. In addition, the toxin is less unstable at neutral alkaline pH; therefore, we chose extreme pH ranges present in the stomach (pH 1.2) and intestinal fluids (pH 7.5) to answer questions about the accessibility of the toxin in vivo.

  1. Guncheva, M.; Stippler, E. Effect of Four Commonly Used Dissolution Media Surfactants on Pancreatin Proteolytic Activity. AAPS PharmSciTech 2017, 18, 1402–1407, doi:10.1208/s12249-016-0618-8.
  2. Pharmacopeia, U. S. National Formulary USP 38—NF 33. In Rockville: The United States Pharmacopeial Convention.

#While the authors mention they used equal volume of culture for the dissolution test, how did they ensure equal culture volume also equates to equal number of IETP cells per condition. Provide details in the method.

Authors' response: The experiments were performed with culture of the strain ITEP-24 with cell density of 3.29 x 106, as described in the text. From the homogenized culture, we perform dilutions for dissolution experiments. Therefore, we believe that the distribution of cells among the test solutions has been the same as checked by microscopy and cell counting.

Reviewer 3 Report

This work does not provide any important contribution to the field of interest of the research. Besides, there are some important errors that the authors should take into consideration to reedit this work (some of them are compiled below). It is also necessary a deep revision of English style, as sometimes it is difficult to understand what the authors are trying to explain.

Some aspects to correct:

  1. 89-99: Figure 1 is not coherent with information in Tables S1 and S2, neither with this paragraph. If the information is that in Figure 1, then this paragraph and tables should be reordered.

  1. 115-117: it is not understandable the information about those four times

First and second paragraphs of “Discussion section”:  the authors should state clearly the aim of this work. The last sentence of second paragraph is badly written to be understood.

Author Response

São Paulo, October 30th, 2020

Authors’ responses - Referrers – Pharmaceuticals

Manuscript: “Availability of Guanitoxin in water samples containing Sphaerospermopsis torques-reginae cells submitted to dissolution tests”.

Open Review 3

Comments and Suggestions for Authors

This work does not provide any important contribution to the field of interest of the research. Besides, there are some important errors that the authors should take into consideration to reedit this work (some of them are compiled below). It is also necessary a deep revision of English style, as sometimes it is difficult to understand what the authors are trying to explain.

Some aspects to correct:

Authors' response: Thanks for the comment, but we disagree with your opinion. Our work was developed considering a global environmental problem, "accidental poisoning of wild and domestic animals by guanitoxin-producing cyanobacteria." Our work is original and well bring new data, showing the relative amount of toxin released by cells of the ITEP-24 strain in a simulated gastrointestinal tract medium. Therefore, we believe that our study can help in the diagnosis of diseases caused by cyanotoxins, as well as in the appropriate treatment of animals intoxicated by guanitoxin, thus being able to prevent the death of many organisms.

#89-99: Figure 1 is not coherent with information in Tables S1 and S2, neither with this paragraph. If the information is that in Figure 1, then this paragraph and tables should be reordered.

Authors' response: Thanks for the comment, we agreed and have made the correction in the text to be clearer.

Figure 1 is a representation of the raw data obtained in LC-MS. The data present in tables S1 and S2 also refer to the raw data obtained in the analyzes in LC-MS. In the data shown in Figure 2, it refers to the normalized concentration in percentage (%), this is the most common way to display graphs showing dissolution curves.

#115-117: it is not understandable the information about those four times.

Authors' response: Thanks for the comment; we have rewritten this sentence to make it understandable.

#First and second paragraphs of “Discussion section”:  the authors should state clearly the aim of this work.

Authors' response: we have added the purpose of the study in the second paragraph of the discussion

# the last sentence of second paragraph is badly written to be understood.

Authors' response: Thanks for the comment, we have rewritten this sentence.

Round 2

Reviewer 1 Report

The authors have improved the articles according to my suggestions. In my opinion, the article is now suitable for publication.

Reviewer 2 Report

All the comments have been addressed and the manuscript is now deemed fit for publication.

Reviewer 3 Report

The manuscript has improved remarkable in its actual state, therefore it is now publishable

This manuscript is a resubmission of an earlier submission. The following is a list of the peer review reports and author responses from that submission.